# Detection of Genotype-Specific Antibody Responses to Glycoproteins B and H in Primary and Non-Primary Human Cytomegalovirus Infections by Peptide-Based ELISA

**DOI:** 10.3390/v13030399

**Published:** 2021-03-03

**Authors:** Federica Zavaglio, Loretta Fiorina, Nicolás M. Suárez, Chiara Fornara, Marica De Cicco, Daniela Cirasola, Andrew J. Davison, Giuseppe Gerna, Daniele Lilleri

**Affiliations:** 1Laboratorio Genetica—Trapiantologia e Malattie Cardiovascolari, Fondazione IRCCS Policlinico San Matteo, 27100 Pavia, Italy; fede.zavaglio90@gmail.com (F.Z.); loretta.fiorina@gmail.com (L.F.); c.fornara@smatteo.pv.it (C.F.); maricadecicco23@gmail.com (M.D.C.); daniela.cirasola@libero.it (D.C.); g.gerna@smatteo.pv.it (G.G.); 2Microbiologia e Virologia, Fondazione IRCCS Policlinico San Matteo, 27100 Pavia, Italy; 3Laboratorio Biochimica-Biotecnologie e Diagnostica Avanzata, Fondazione IRCCS Policlinico San Matteo, 27100 Pavia, Italy; 4MRC-University of Glasgow Centre for Virus Research, Glasgow G61 1AF, UK; Nicolas.Suarez@glasgow.ac.uk (N.M.S.); andrew.davison@glasgow.ac.uk (A.J.D.)

**Keywords:** human cytomegalovirus, glycoprotein B, glycoprotein H, genotype, antibody, primary infection, non-primary infection

## Abstract

Background: Strain-specific antibodies to human cytomegalovirus (HCMV) glycoproteins B and H (gB and gH) have been proposed as a potential diagnostic tool for identifying reinfection. We investigated genotype-specific IgG antibody responses in parallel with defining the gB and gH genotypes of the infecting viral strains. Methods: Subjects with primary (*n* = 20) or non-primary (*n* = 25) HCMV infection were studied. The seven gB (gB1-7) and two gH (gH1-2) genotypes were determined by real-time PCR and whole viral genome sequencing, and genotype-specific IgG antibodies were measured by a peptide-based enzyme-linked immunosorbent assay (ELISA). Results: Among subjects with primary infection, 73% (*n* = 8) infected by gB1-HCMV and 63% (*n* = 5) infected by gB2/3-HCMV had genotype-specific IgG antibodies to gB (gB2 and gB3 are similar in the region tested). Peptides from the rarer gB4-gB7 genotypes had nonspecific antibody responses. All subjects infected by gH1-HCMV and 86% (*n* = 6) infected by gH2-HCMV developed genotype-specific responses. Among women with non-primary infection, gB and gH genotype-specific IgG antibodies were detected in 40% (*n* = 10) and 80% (*n* = 20) of subjects, respectively. Conclusions: Peptide-based ELISA is capable of detecting primary genotype-specific IgG responses to HCMV gB and gH, and could be adopted for identifying reinfections. However, about half of the subjects did not have genotype-specific IgG antibodies to gB.

## 1. Introduction

Human cytomegalovirus (HCMV) is a major cause of congenital infection and non-hereditary sensorineural hearing loss [1]. Congenital infection occurs in 30–40% cases after primary infection during pregnancy [2,3,4], and has also been documented in HCMV-immune mothers [5] as a consequence of non-primary infection. Whether transmission during non-primary infection is the consequence of reinfection or reactivation is debatable, but this information would be relevant in designing prevention strategies.

To distinguish whether a non-primary HCMV infection is due to reinfection or reactivation, it would be necessary to compare the genome of the virus present during non-primary infection with that responsible for primary infection. Strain-specific antibody responses have been investigated as a potential alternative diagnostic tool for identifying reinfection. By utilizing the known heterogeneity within the epitopes on envelope glycoproteins H (gH) and B (gB) of HCMV strains AD169 and Towne [6,7], a peptide-based enzyme-linked immunosorbent assay (ELISA) was developed to detect serological responses to infection with different HCMV strains [8,9]. Women who delivered congenitally infected infants exhibited evidence of infection with multiple HCMV strains, suggesting that maternal reinfection after exposure to different virus strains is a risk factor for delivery of a congenitally infected infant [8,10,11].

The objective of this study was to investigate the development of genotype-specific IgG antibody responses to gB and gH during primary infection in parallel with genotyping the corresponding genes in the virus strain responsible for the infection. To design a genotype-specific peptide-based ELISA, we used the information available for large numbers of gB and gH sequences [12].

## 2. Materials and Methods

### 2.1. Study Subjects

The study enrolled 20 subjects (19 pregnant women and one male) with primary HCMV infection. Sequential samples (blood, urine, saliva and, for women, vaginal swabs) were collected 1–24 months after onset of infection. Samples from the same sources and breast milk were collected 2–3 months after delivery from 25 women with non-primary infection. Subjects with primary infection were enrolled at the Obstetrics and Gynecology clinics (pregnant women) or at the Microbiology and Virology unit (a non-pregnant subject). Diagnosis and dating of primary HCMV infection was achieved based on two or more of the following criteria, as previously reported [13]: (i) appearance of HCMV-related symptoms as well as biochemical and hematological signs associated with HCMV infection; (ii) IgG seroconversion, (iii) seroconversion of neutralizing antibodies (Nt), which occurs 4-6 weeks after onset of primary infection in human fibroblast cell cultures [14], (iv) kinetics of HCMV-specific IgM and IgG antibodies, (v) low IgG AI, and (vi) presence of HCMV DNA in blood [14]. Women with non-primary infection were enrolled at the Obstetrics and Gynecology clinics. Non-primary infection was defined as the detection of HCMV DNA in bodily fluids after delivery in a woman with preconception HCMV-specific IgG antibody. Twenty HCMV-seronegative non-pregnant subjects (13 male and 7 female) were enrolled as controls. All the subjects signed a written informed consent.

### 2.2. Extraction, Quantification and Real-Time PCR-Based Genotyping of HCMV DNA

DNA was extracted from urine, breast milk and vaginal and saliva swabs using an EZ1 DSP virus kit (Qiagen, Hilden, Germany), and from blood using a QIAmp DNA mini kit (Qiagen). HCMV DNA was quantified using an Artus CMV RG PCR kit (Qiagen). Four of the seven gB genotypes (gB1-4) and the two gH genotypes (gH1-2) were genotyped by PCR [15]. Genotyping of gB (gB1, gB2, gB3, and gB4) and gH (gH1 and gH2) was performed using two multiplex real-time PCR assays. Primers and probes specific for each genotype were designed at the N terminus of gB and gH except for gB3, for which the cleavage region was amplified. However, in this study gB2 and gB3 were considered collectively as gB2/3. Genotypes gB5-7 are not detectable by this PCR assay. The primer and probe sequences were as described previously [15]. Each multiplex assay contained 10 µL DNA extract, 25 µL HotStar Master mix (Qiagen, Hilden, Germany), 0.3 µM primers, 0.2 µM probes, and 4.5 mM MgCl_2_. Template denaturation and activation of HotStarTaq DNA polymerase were achieved by heating for 15 min at 95°C. This was followed by 45 cycles of denaturation at 95°C for 30 s, annealing at 55°C for 30 s, and extension at 72 °C for 30 s. The assay was carried out in a Rotor-Gene Q 5PLEX instrument (Qiagen, Hilden, Germany).

### 2.3. Whole-Genome Sequencing Based Genotyping

In six subjects with primary infection, HCMV whole-genome sequencing was conducted directly from clinical samples as previously described [12]. Briefly, an aliquot of 50 µL of extracted DNA was fragmented using an LE220 sonicator (Covaris, Woburn, MA, USA) at an average fragment size of 500 bp. Sequencing libraries were prepared following a target enrichment approach (SureSelect^XT^ version 1.7, Agilent Technologies, Santa Clara, CA, USA). HCMV-enriched libraries were loaded on a NextSeq DNA sequencer (Illumina, San Diego, CA, USA) to generate 2 × 150 bp paired-ended reads. Sequencing reads were assembled using SPAdes 3.5.0 [16]. The sequences corresponding to gB and gH were extracted from the assembled contigs.

### 2.4. Peptide-Based ELISA

An ELISA was used to detect peptide-specific IgG antibodies capable of recognizing linear peptides near the N-termini of the proteins encoded by the gB1-7 and gH1-2 genotypes [12,17].

Half-area 96-well microplates were coated for 1 h with streptavidin (0.1 M) diluted in bicarbonate buffer. After overnight (or 1 h) blocking with 5% (wt/vol) skimmed milk, biotinylated peptides (170 nM) were immobilized in the wells for 1 h. After washing, the plates were incubated for 1 h with human serum (diluted 1:50), then for 45 min with horseradish peroxidase-labeled goat IgG to human IgG and, finally, for 25 min with 30 mg/mL orthophenylendiamine before the addition of 4 N sulphuric acid. The optical density (OD) value of the serum incubated without peptide was subtracted from the OD value of the serum incubated with peptide. Mean OD values were calculated from triplicate assays.

## 3. Results

### 3.1. Selection of gB and gH Peptides for Detecting Genotype-Specific Antibodies

The most variable regions in gB are located near the N terminus of the predicted primary translation product (after the signal peptide cleavage site) and at the proteolytic cleavage site located centrally in the protein. The presence of strain-specific epitopes has been reported for the former [6] but the latter is cleaved in the mature protein and is not an antibody-binding site [18]. On the basis of the genome sequences of 244 HCMV strains [12], the N-terminal region was found to provide the greater specificity among genotypes (Figure 1A and Figure 2). However, genotypes 2 and 3 cannot be distinguished in this region and were considered collectively (gB2/3). This region encompasses amino acid residues (aa) 25–72 of gB1 (the numbering refers to the predicted primary translation product) and gB2/3, aa 25–70 of gB4, aa 25–74 of gB5, aa 25–73 of gB6, and aa 25–71 of gB7. Within the 244 genomes, 39% of gB sequences belong to gB1, 45% to gB2/3, 11% to gB4, and the remaining 5% to gB5, gB6 or gB7 [12]. Four peptides of 20 aa overlapping by 10–13 aa were designed for each genotype for use in the peptide-based ELISA (see Figure 1A for peptide names and sequences). Strains of genotype gB2/3 showed either an alanine or a threonine in position 34 and either a tyrosine or a serine in position 53. Therefore, a mixture of peptides containing both variants was used.

The region consisting of aa 15–43 of gH1 and aa 14–42 of gH2 [7,8] was selected on the basis of 255 genome sequences (56% gH1 and 44% gH2) for detecting genotype-specific antibodies (Figure 1B), and there are no other major variable regions in gH. Two peptides of 20 aa overlapping by 11 aa were designed for each gH genotype for use in the peptide-based ELISA (Figure 1B).

### 3.2. Genotypes of gB and gH in Subjects with Primary Infection

Clinical samples collected from subjects with primary infection were genotyped for HCMV gB and gH by PCR or whole-genome sequencing (Table 1). Quantitative levels of HCMV-DNA detected in the clinical samples analyzed are reported in Table 2. A single genotype was detected consistently throughout the entire follow-up period and in different bodily fluids from each subject. HCMV whole-genome sequencing was performed for five subjects and confirmed the real-time PCR-based genotyping results, and in one additional subject detected the gB7 genotype. The HCMV genome sequences for four subjects (P-083, P-225, P-226, and P-232) were deposited in GenBank under accession numbers MW197154, MW197155, MW197156, and MW197157, respectively.

### 3.3. Genotype-Specific Peptide-Based ELISA in Seronegative Controls and Subjects with Primary HCMV Infection

To determine cutoff OD values for the peptide-based ELISA, samples from 20 HCMV-seronegative subjects were tested. A value of ≤0.1 (mean + three standard deviations of OD values detected in seronegative subjects) was considered negative, and a value of ≥0.3 (three times above the cutoff value) was considered positive. Intermediate values were considered to be undetermined and were counted as negative.

Sequential serum samples from 20 subjects with primary HCMV infection were then tested. Among the 11 subjects infected by gB1-HCMV, eight showed an IgG response to peptides of the homologous genotype (Table 1, Figure 3A and Figure 4). Specifically, five developed an IgG response to peptide gB1.2 (four of these also to gB1.3), and three to peptide gB1.4 (two of these also to gB1.3). No response was detected to peptide gB1.1. Some subjects showed genotypic cross-reactivity, in particular to gB5 and gB6 and, to a lesser extent, gB4 and gB7. No subjects infected by gB1 showed a response to gB2/3.

Of the eight subjects infected by gB2/3-HCMV, five developed an IgG response to peptides of the homologous genotype (Table 1, Figure 3B and Figure 4). Specifically, three developed an IgG response to peptide gB2/3.2 (one of these lost this response at later times), one to gB2/3.4 and one to both peptides. No subject reacted with gB2/3.1 or gB2/3.3. Some subjects showed a cross-reactivity to gB4 and gB7 and, to a lesser extent, gB6. No subject infected by gB2/3-HCMV showed a response to gB1 or gB5.

The only pregnant woman presenting gB7-HCMV had no antibody response against peptides designed for gB7 or any other genotype.

In summary, peptides derived from gB1 and gB2/3 detected a mutually exclusive genotype-specific IgG response, whereas peptides derived from gB4, gB5, gB6, and gB7 detected only nonspecific responses.

Regarding the IgG response to gH, 13 subjects infected by gH1-HCMV developed an IgG response to peptide gH1.2 but not gH1.1, whereas six subjects infected by gH2-HCMV developed an IgG response to gH2.2 but not gH2.1. No subject showed cross-reactivity to the heterologous gH genotype (Figure 5A,B).

Overall, 8/11 (73%) subjects infected by gB1-HCMV, 5/8 (63%) of subjects infected by gB2/3-HCMV (Figure 6A), 13/13 (100%) of subjects infected by gH1-HCMV, and 6/7 (86%) of subjects infected by gH2-HCMV (Figure 6B) showed a genotype-specific IgG response. Genotype-specific IgG responses to gH developed earlier than those to gB (median time: 115 vs. 365 d, *p* = 0.02) and in a greater proportion of subjects (95% vs. 65%, Figure 6C). The individual follow-up of five representative subjects is shown in Figure 7.

### 3.4. PCR-Genotyping and Genotype-Specific Peptide-Based ELISA in Subjects with Non-Primary HCMV Infection

Genotype-specific IgG responses to gB1, gB2/3, gB4, gH1, and gH2 were tested in 25 women with non-primary HCMV infection (Table 1). A multiple-genotype infection was detected in three (12%) subjects. Peptide-based ELISA detected gB and gH genotype-specific IgG in 10 (40%) and 20 (80%) subjects, respectively. Multiple genotype-specific IgG responses were detected in one of the three subjects with PCR-diagnosed multiple infection, and in one subject with a single genotype detected by PCR.

Among the 23 subjects in whom a single gB genotype was detected (11 gB1-HCMV, 11 gB2/3-HCMV and one gB4-HCMV), nine (39%) showed a homologous IgG response, one (4%) showed an apparently multiple response (the response to both gB2/3 and gB4 was not considered to be multiple, since gB4 peptides were cross-reactive with gB2/3 peptides), and 13 (57%) showed no IgG response. No response was observed in the two subjects with multiple gB genotypes (Figure 8A). Among the 24 subjects in whom a single gH genotype was detected, 16 (67%) showed a homologous IgG response, five (21%) showed no response, and three (12%) showed a heterologous response. One subject with multiple gH genotypes showed also a multiple IgG antibody response (Figure 8B). In primary HCMV infections we observed a complete overlap between the HCMV genotype detected and the IgG antibody response. Conversely, in non-primary infections, some heterologous responses with respect to genotype were observed (i.e., anti-gH1 antibody was detected in two subjects with gH2-HCMV DNA, and anti-gH2 antibody was detected in a subject with gH1-HCMV DNA in bodily fluids).

## 4. Discussion

This study investigated the development of the IgG antibody response specific for gB and gH genotypes in subjects with HCMV primary or non-primary infection for whom the gB and gH genotypes of the infecting HCMV strains were known. The major findings indicate that linear epitopes near the N-terminus of gB1 and gB2/3 (which represent the great majority of circulating strains) elicit a mutually exclusive genotype-specific IgG response. However, there was some level of cross-reactivity between antibodies elicited by gB1 with those elicited by gB5 and gB6, and antibodies elicited by gB2/3 with those elicited by gB4 and gB7. Similarly, linear epitopes at the N-terminus of gH also elicited mutually exclusive genotype-specific IgG responses. Genotype-specific IgG responses to gB appeared late after primary infection (median time of 12 m), were not detectable in all subjects, and could become undetectable with time (especially those to gB2/3). In contrast, genotype-specific anti-gH IgG antibody appeared sooner (median time of 4 m) and was detected in almost all subjects.

Similarly, in the 25 postpartum women developing a non-primary infection, genotype-specific anti-gB IgG was not detected in about half of the cases, whereas type-specific anti-gH antibodies were detected in the great majority of cases. With a few exceptions, during non-primary infection the IgG specificity matched the genotype of the HCMV strain detected. The exceptions showed either a heterologous IgG response or a multiple IgG response in the presence of a single gB or gH genotype. It is possible that the genotype-specific IgG represented the footprint of a previous encounter with a different HCMV strain, whose genome was not detected during the non-primary infection episode, either because it was not replicating or because it represented an undetected minor component (in this case whole-genome sequencing would help). In this study it is impossible to determine whether non-primary infections represent reactivations or reinfections. Since viral DNA has been detected for the most part in the breast milk, they are more likely reactivations. This would be consistent with the correlation between genotype and serotype observed in most cases. However, a limitation of this study is the lack of longitudinal serological and virological testing using preconception and postdelivery samples for discrimination between maternal reactivation or reinfection during pregnancy.

The presence of polymorphic strain-specific linear epitopes on gB and gH was documented in the 1990s [6,7], using polypeptides from the AD169 (gB2/3-gH1) and Towne (gB1-gH2) strains. Subsequently, two assays were developed to detect strain-specific IgG with the purpose of diagnosing reinfections with different HCMV strains in immunocompetent women [8,9,10] and transplant recipients [19]. By comparing HCMV genotyping and serological results, we can now define the strain-specific peptide-based ELISA as a genotype-specific peptide-based ELISA: the AD169-like response is the gB2/3- and gH1-specific response, and the Towne-like response is the gB1- and gH2-specific response. Moreover, due to the overlapping responses observed with peptides from gB1, gB5 and gB6 on one side, and gB2/3, gB4 and gB7 on the other side, our data suggests that the genotype-specific antibody response to gB could actually be divided in two groups. The first (gB1, gB5 and gB6) represents the AD169-like response and the second (gB2/3, gB4 and gB7) represents the Towne-like response.

A limitation of the assays was that about 25–40% of the subjects tested did not show reactivity with any antigen. This was interpreted as the consequence of either sensitivity to low levels of strain-specific antibodies or infection with viruses containing gH or gB epitope variants that were not represented among the antigens used. On the basis of the available genomic data [12], the presence of other epitopic variants on gH is less plausible, and only two genotypes have been identified in which diversity is limited to the N-terminal region [17]. The diversity of gB is more complex, also because of the occurrence of recombination events within the gB gene. Seven genotypes have been described based on the variability in the N-terminal and cleavage site regions [12,20]. However, variability outside these two regions is low and therefore less likely to generate polymorphic epitopes (see Figure 2), and the cleavage site region is not recognized as an antibody-binding site. This restricts the potential presence of polymorphic epitopes to the N-terminus, in which we can hypothesize that at least two distinct polymorphic epitopes exist in both gB1 and gB2/3, since two nonoverlapping peptides reacted with serum samples. We hypothesize that the lack of reactivity observed in our and the previously described peptide-based ELISA indicates that not all subjects develop detectable IgG to the polymorphic epitopes near the N-terminus of gB, whereas the existence of other epitope variants not represented in the assays could be excluded. In accord with this hypothesis, a previous study showed that only about 50% of subjects developed antibodies against the nonpolymorphic region in the N-terminus [21,22], which is immediately downstream from the polymorphic peptides used in our study. A limitation of the study was that we were unable to determine the IgG response elicited by gB4-gB7 HCMV strains, because these are much less frequent in the population. In particular, genotypes gB5, gB6 or gB7 represent a negligible proportion of circulating strains that could be rarely observed in the clinical practice (less than 2% of the sequenced strains). Therefore, it is unlikely to detect infections caused by strains belonging these genotypes, unless hundreds of cases are collected. Further study of infections caused by these rare genotypes would define whether these strains can elicit antibodies against the polymorphic N-terminus and to which extent they cross-react with the epitopes on gB1 and gB2/3.

Owing to a study design based on the large number of HCMV genome sequences available, and to the validation performed on subjects infected by known gB and gH genotypes, the genotype-specific peptide-based ELISA described in this study appears to be a valuable tool to investigate the extent and clinical impact of reinfections. Some studies have argued that maternal superinfection with a serologically distinct HCMV strain may play a significant role in congenital HCMV infection [8,11]. Follow-up of HCMV-seropositive subjects at high risk of reinfection due to exposure to multiple strains (i.e., mothers of children attending day-care centers [23]) by parallel virological, genomic and serological monitoring would provide an opportunity for validating the serological diagnosis of reinfections with different virus strains. HCMV whole-genome sequence analysis would be the gold standard for diagnosis of reinfections, but for this approach a degree of lytic infection greater than that required for PCR analysis is probably necessary. The major limitation of the peptide-based ELISA is the lack of detection of genotype-specific antibodies to gB in about 30–50% subjects. However, genotype-specific anti-gH IgG was detected more frequently. In future, the discovery of potential genotype-specific polymorphic epitopes on other more variable envelope glycoproteins, such as glycoproteins N and O [24], may improve the serological identification of HCMV strains.

## Figures and Tables

**Figure 1 viruses-13-00399-f001:**
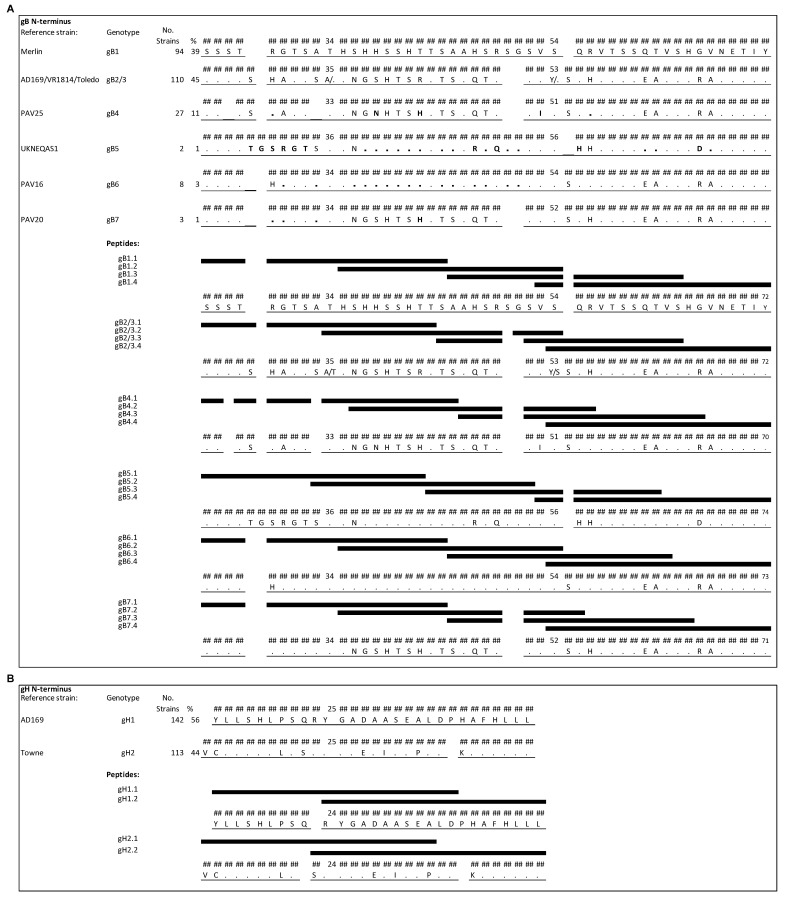
N-terminal regions of gB and gH genotypes aligned to genotype-specific peptides (black bars). Numbering is relative to the primary translation product, with the signal peptide present. (**A**) N-terminal region of gB showing the greatest diversity among the seven genotypes (gB1-7). The amino acid sequences of HCMV strains representing the genotypes are shown, using the strain Merlin sequence (gB1) as a reference. Amino acid residues (aa) that differ from those in the reference are reported, and residues that are identical in the reference are indicated by dots. Positions in gB4-7 that differ from those in gB2/3 are represented in bold. Each peptide differs in at least one residue except gB4.4 and gB6.4, which are identical, as well as one of the two peptides contained in the mixture of gB2/3.4 and peptide gB7.4. (**B**) N-terminal region of gH that distinguishes the two genotypes (gH1-2). Each peptide was biotinylated at the N-terminus, a flexible hydrophilic linker (trioxatridecan-succinamic acid) was inserted between the biotin moiety and the peptide, and a glycine amide was added at C-terminus.

**Figure 2 viruses-13-00399-f002:**
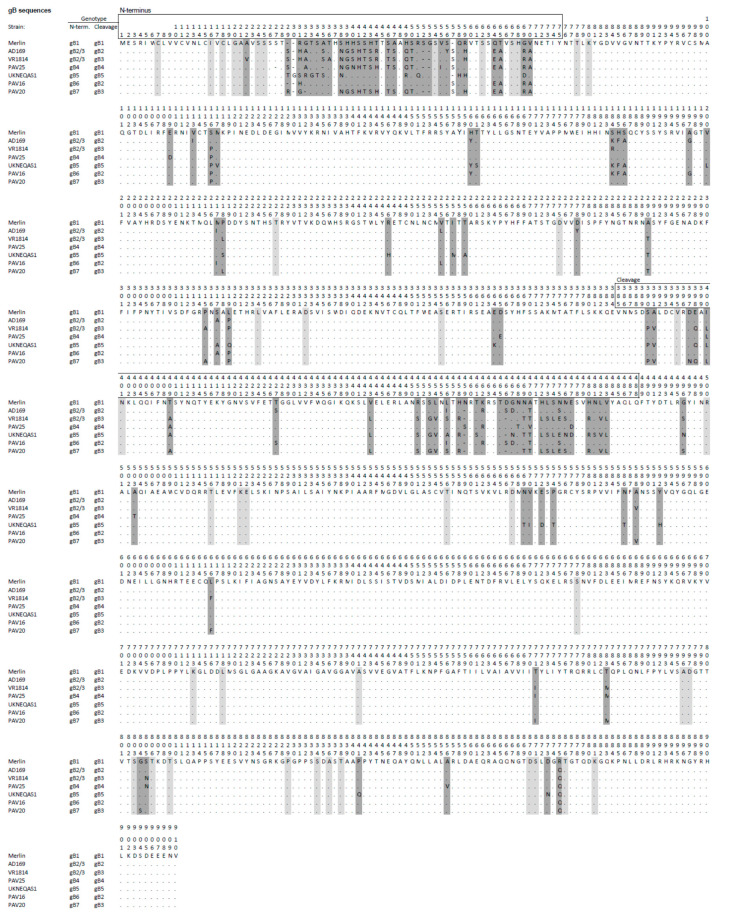
Predicted primary amino acid sequences representing the seven gB genotypes (gB1-7) in HCMV strains, with the strain Merlin sequence (gB1) shown as the reference. Amino acid residues that differ from those in the reference are reported, residues that are identical to those in the reference are indicated by dots, and gaps are shown by hyphens. Dark grey shading indicates major differences that characterize the genotype, and light grey shading denotes differences observed in individual strains. The numbers (1–910) above the sequences indicate residue numbering. Genotype classifications at the N terminus and cleavage regions are reported to the left of the sequences.

**Figure 3 viruses-13-00399-f003:**
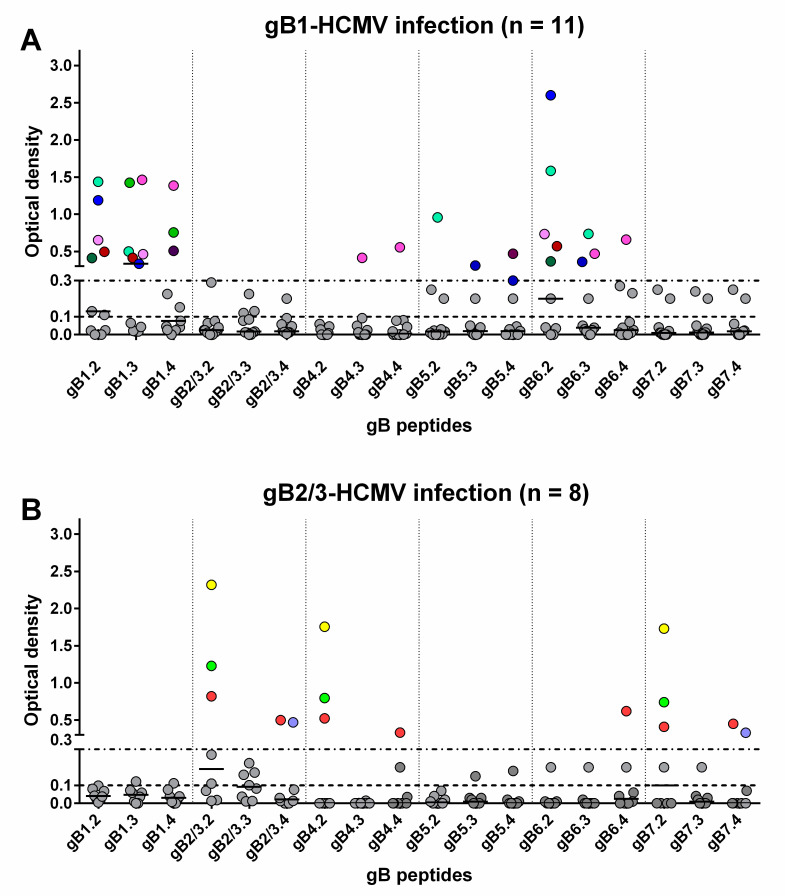
Genotype-specific IgG antibody (OD value) to peptides of the seven gB genotypes 1 year after primary HCMV infection. Each subject is represented by points in the same color above an OD value of 0.3. (**A**) IgG reactivity in subjects infected by gB1-HCMV. (**B**) IgG reactivity in subjects infected by gB2/3-HCMV.

**Figure 4 viruses-13-00399-f004:**
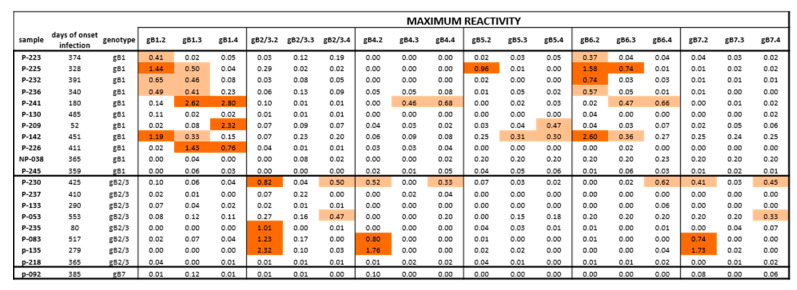
Maximum IgG antibody reactivity (OD value) against peptides of different gB genotypes in subjects with HCMV primary infection. Positive responses are highlighted in light orange for low reactivity (<0.70 OD) or dark orange for high reactivity (>0.70 OD). Some subjects infected by gB1 showed a cross-reactivity in particular with gB5 and gB6 and no response to gB2/3. Subjects infected by gB2/3 showed a cross reactivity to gB4 and gB7 and no response to gB1 and gB5. The only pregnant women typed as gB7 had no antibody response against gB7 or other genotypes.

**Figure 5 viruses-13-00399-f005:**
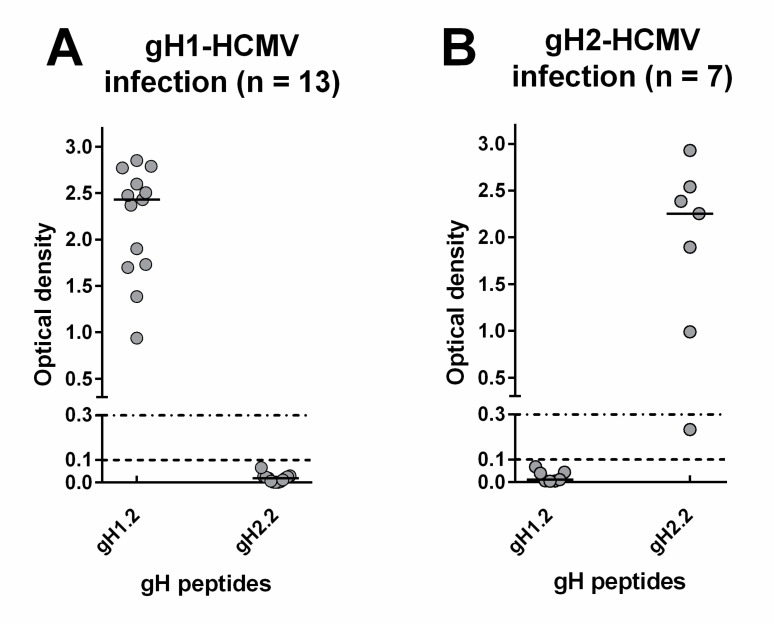
Genotype-specific IgG antibody (OD value) to peptides of the two gH genotypes 1 year after primary HCMV infection. (**A**) IgG response in subjects infected by gH1-HCMV. (**B**) IgG response in subjects infected by gH2-HCMV.

**Figure 6 viruses-13-00399-f006:**
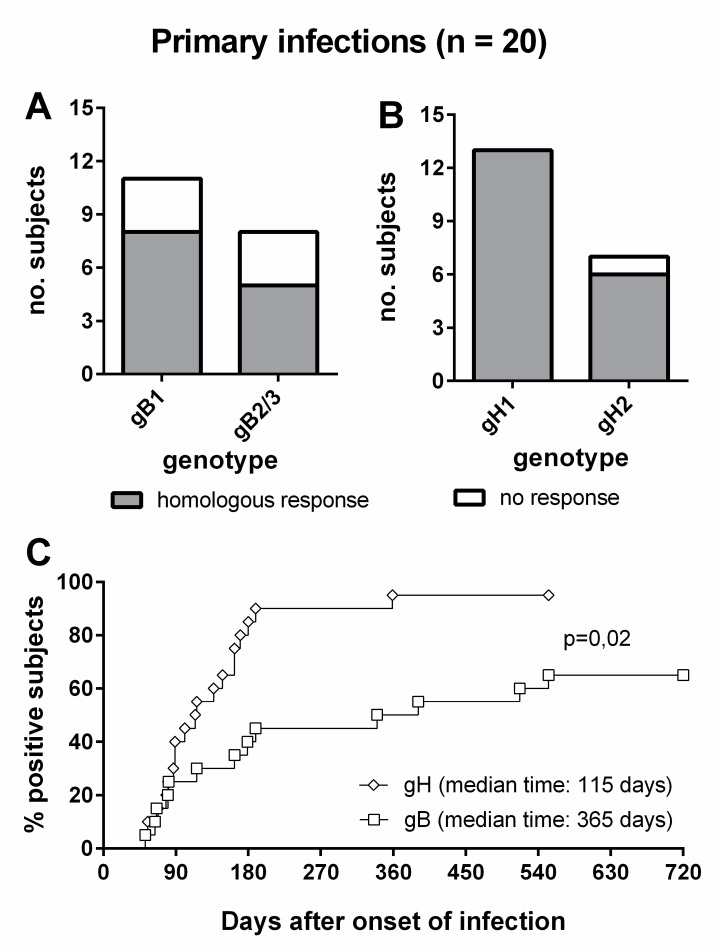
Genotype-specific IgG antibody to gB and gH in subjects with primary HCMV infection. (**A**) gB genotype-specific IgG response in subjects infected by gB1- and gB2/3-HCMV. (**B**) gH genotype-specific IgG response in subjects infected by gH1- and gH2-HCMV. (**C**) Comparative time to development of genotype-specific IgG response to gB and gH.

**Figure 7 viruses-13-00399-f007:**
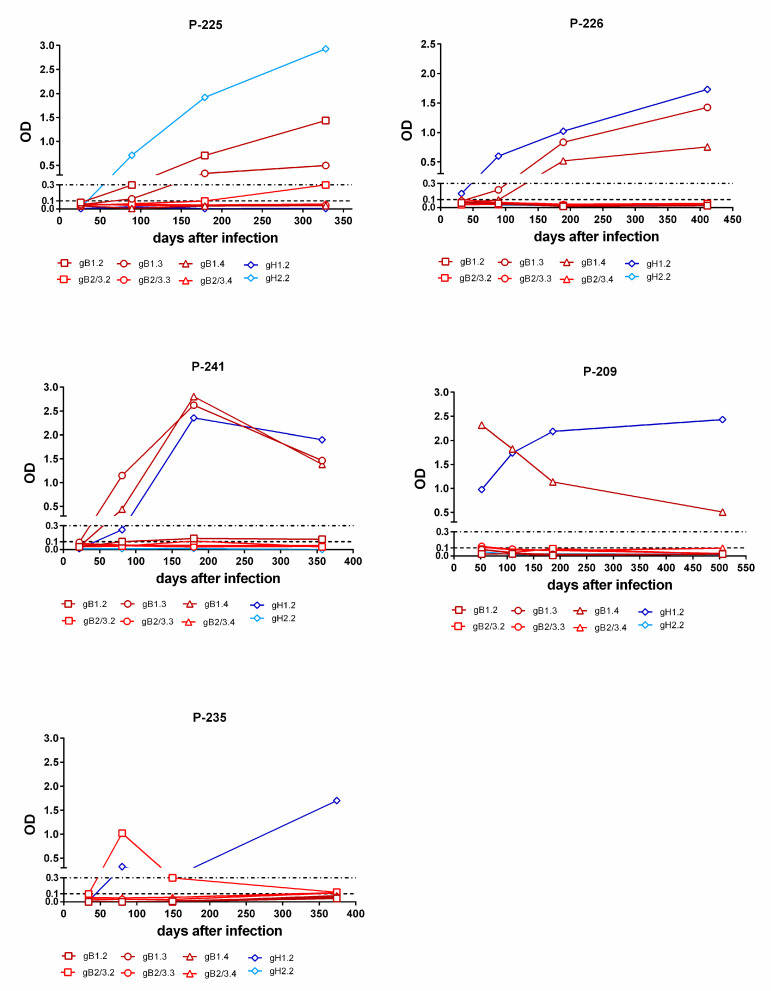
Kinetics of genotype-specific IgG antibody response to peptides representing gB and gH genotypes in single women with primary infection. In subjects P-225 and P-226, the gB and gH IgG responses increases during the first year after infection. In subject P-241, the gB and gH IgG responses decreased over time. In subject P-209, the gB IgG response decreased and the gH IgG response remained constant with time. Finally, in subject P-235, the gH IgG response increased at late times (after 6 m) and the initial gB IgG response disappeared.

**Figure 8 viruses-13-00399-f008:**
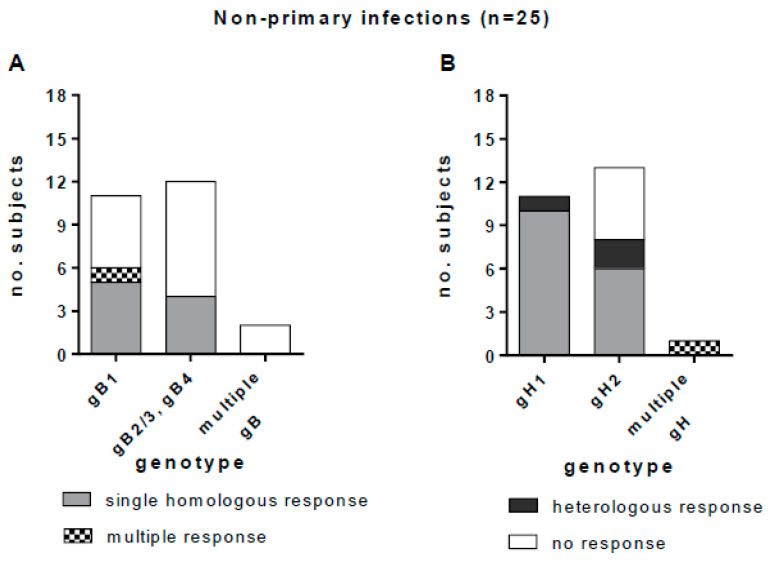
Genotype-specific IgG antibody to gB and gH in subjects with non-primary HCMV infection. (**A**) gB genotype-specific IgG response in subjects infected by gB1-HCMV, gB2/3-HCMV or gB4-HCMV or multiple genotype HCMV strains (gB2/3 and gB4 are grouped together, since the antibody response to these genotypes was cross-reactive). (**B**) gH genotype-specific IgG response in women infected by gH1-HCMV, gH2-HCMV or multiple genotype HCMV strains.

**Table 1 viruses-13-00399-t001:** gB and gH genotypes and serum antibody types of 20 subjects with primary infection and 25 women with non-primary infection.

Subject	Specimen (Days after Infection)	HCMV Genotype	Serum Antibody Type
PCR-Genotyping	Whole HCMV Sequencing	gB	gH	gB	gH
**Primary infection**
NP-038	U (55), U (173)		gB1	gH1	n.d.	gH1
P-130	S (87)		gB1	gH1	n.d.	gH1
P-209 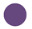	U (52)		gB1	gH1	gB1 (gB5)	gH1
P-223 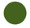	U (21), S (374)		gB1	gH1	gB1 (gB6)	gH1
P-226 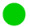	U (32), U (189)	V (189)	gB1	gH1	gB1 (gB5, gB6, gB7)	gH1
P-232 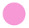	V (92), V (194), AF (120)	V (92), AF (120)	gB1	gH1	gB1 (gB6)	gH1
P-241 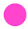	V (54), U (357)		gB1	gH1	gB1 (gB4, gB6)	gH1
P-245	V (77), U 359)		gB1	gH1	n.d.	gH1
P-142 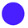	U (38)		gB1	gH2	gB1 (gB5, gB6)	gH2
P-225 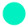	V (26), S (328), V (328)	V (26)	gB1	gH2	gB1 (gB5, gB6)	gH2
P-236 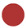	V (85), U (340)		gB1	gH2	gB1 (gB6)	gH2
P-083 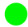	S (21)	B (21)	gB2/3	gH1	gB2/3 (gB4, gB7)	gH1
P-218	S (59)	S (21)	gB2/3	gH1	n.d.	gH1
P-235	V (34), U (374)		gB2/3	gH1	gB2/3	gH1
P-237	V (100), V (410)		gB2/3	gH1	n.d.	gH1
P-053 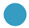	U (43)		gB2/3	gH2	gB2/3 (gB7)	n.d.
P-133	S (137), U (137), U (178)		gB2/3	gH2	n.d.	gH2
P-135 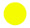	U (152)		gB2/3	gH2	gB2/3 (gB4, gB7)	gH2
P-230 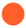	V (64), S (83)		gB2/3	gH2	gB2/3 (gB4, gB6, gB7)	gH2
P-092		AF (85)	gB7	gH1	n.d.	gH1
**Non-primary infection**
WP1-061	M, S		gB1	gH1	gB1	gH1
WP1-115	M		gB1	gH1	n.d.	gH1
WP1-085	M		gB1	gH1	gB1	gH1
WP1-122	M		gB1	gH1	gB1, **gB2/3 (gB4)**	gH1
WP1-065	M		gB1	gH2	gB1	n.d.
WP1-125	M		gB1	gH2	gB1	n.d.
WP1-160	M		gB1	gH2	gB1	gH2
WP1-094	M		gB1	gH2	n.d.	gH2
WP1-111	M		gB1	gH1	n.d.	gH1
WP1-066	S, V		gB1	gH2	n.d.	**gH1**
WP1-055	M		gB1	gH1, gH2	n.d.	gH1, gH2
WP1-056	M		gB2/3	gH1	gB2/3 (gB4)	gH1
WP1-144	M		gB2/3	gH1	gB2/3	gH1
WP1-142	M		gB2/3	gH2	n.d.	gH2
WP1-129	M		gB2/3	gH2	n.d.	gH2
WP1-158	U		gB2/3	gH2	n.d.	**gH1**
WP1-139	M		gB2/3	gH2	gB2/3 (gB4)	n.d.
WP1-154	M		gB2/3	gH2	n.d.	gH2
WP1-099	S, U		gB2/3	gH1	n.d.	gH1
WP1-120	U		gB2/3	gH2	n.d.	gH2
WP1-058	M		gB2/3	gH2	n.d.	n.d.
WP1-163	V, S		gB2/3	gH1	n.d.	gH1
WP1-075	M		gB4	gH2	gB2/3 (gB4)	n.d.
WP1-162	V		gB2/3, gB4	gH1	n.d.	gH1
WP1-119	M		gB2/3, gB4	gH1	n.d.	**gH2**

n.d., not detected; AF, amniotic fluid; B, blood; M, milk; S, saliva swab; U, urine; V, vaginal swab. Samples from women with non-primary infection were collected postpartum. When detected, serum antibody reactivity to gB4-7 is indicated in brackets because it was found to be nonspecific but cross-reactive. Bold characters indicate heterologous responses with respect to the genotype detected. Colored dots indicate patients represented in Figure 3, who maintain genotype-specific IgG antibody response 1 year after infection (p235 is not represented because loses the IgG antibody response).

**Table 2 viruses-13-00399-t002:** Quantitative levels of HCMV-DNA detected in the clinical samples of 20 subjects with primary infection and 25 women with non-primary infection.

**Subject**	**Specimen (Days after Infection), Copies/mL**		
**Primary infection**			
NP-038	U (55), 426,048	U (173), 7382	
P-130	S (87), 1085		
P-209	U (52), 11,356		
P-223	U (21), 160	S (374), 3845	
P-226	U (32), 660	U (189), 568	V (189), 295
P-232	V (92), 27,923	V (194), 13,887	AF (120), 262,000
P-241	V (54), 19,124	U (357), 2631	
P-245	V (77), 1141	U (359), 592	
P-142	U (38), 894		
P-225	V (26), 192,469	S (328), 429	V (328), 371
P-236	V (85), 15,135	U (340), 3652	
P-083	S (21), 3692	B (21), 6762	
P-218	S (59), 12,078		
P-235	V (34), 559	U (374), 1676	
P-237	V (100), 24,902	V (410), 692	
P-053	U (43), 109		
P-133	S (137), 1176	U (137), 3088	U (178), 1485
P-135	U (152), 809		
P-230	V (84), 932	S (84), 480	
P-092	AF (85), 3,476,390		
**Non-primary infection**			
WP1-061	M, 15,300	S, 550	
WP1-115	M, 719		
WP1-085	M, 64		
WP1-122	M, 584		
WP1-065	M, 600		
WP1-125	M, 872		
WP1-160	M, 234		
WP1-094	M, 122,988		
WP1-111	M, 26		
WP1-066	S, 5246	V, 272	
WP1-055	M, 4600		
WP1-056	M, 300		
WP1-144	M, 1272		
WP1-142	M, 3346		
WP1-129	M, 34,435		
WP1-158	U, 1673		
WP1-139	M, 172		
WP1-154	M, 1842		
WP1-099	S, 254	U, 1040	
WP1-120	U, 719		
WP1-058	M, 480		
WP1-163	V, 10,090	S, 175	
WP1-075	M, 70,800		
WP1-162	V, 2088		
WP1-119	M, 6000		

AF, amniotic fluid; M, milk; S, saliva swab; U, urine; V, vaginal swab.

## Data Availability

Not applicable.

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
