# Peer review of "Detection of Genotype-Specific Antibody Responses to Glycoproteins B and H in Primary and Non-Primary Human Cytomegalovirus Infections by Peptide-Based ELISA"

_viruses, 2021, doi:10.3390/v13030399_

Round 1

Reviewer 1 Report

Proposed manuscript is very well  written, scientific studies are  well organized, assays  used in proposed study are very well described and  conclusions based on results are valid.  Current protocols described in proposed manuscript can improve the serological identification of HCMV strains.  

Manuscript  written by Federica Zavaglio 1,2,3, Loretta Fiorina 1,2, Nicolás M Suárez 4 , Chiara Fornara 1,2,3 , Marica De Cicco 1 , 5 Daniela Cirasola 1 , Andrew J Davison 4 , Giuseppe Gerna 1 , Daniele Lilleri 1 present  experimental  study to detect of genotype-specific antibody responses to glycoprotein B and H in primary and non-primary human cytomegalovirus infections by peptide-based ELISA. Currently, strain-specific antibodies to human cytomegalovirus (HCMV) glycopro- 14 teins B and H (gB and gH) have been proposed as a potential diagnostic tool for identifying reinfection. Authors  investigated genotype-specific IgG antibody responses in parallel with defining the gB 16 and gH genotypes of the infecting viral strains.  They analyzed  samples from  subjects with primary (n=20) or 17 non-primary (n=25) HCMV infection. The seven gB (gB1-7) and two gH (gH1-2) 18 genotypes were determined by real-time PCR and whole viral genome sequencing, and geno- 19 type-specific IgG antibodies were measured by ELISA.As a result  peptide-based ELISA is capable of detecting primary genotype-specific IgG responses 27 to HCMV gB and gH, and could be adopted for identifying reinfections. However, about half of the 28 subjects did not have genotype-specific IgG antibodies to gB.   Manuscript is very well written, detailed description of assays used in studies  are presented  clearly   and   convincible in a paper.  Also, authors described very well all limitations of proposed protocols for detection of genotype-specific antibody responses.   Studies are based on investigation of the development of genotype-specific 51 IgG antibody responses to gB and gH during primary infection in parallel with geno- 52 typing the corresponding genes in the virus strain responsible for the infection.  Authors used the information  from literature available for large 54 numbers of gB and gH sequences to design 53 a genotype-specific peptide-based ELISA,. All results presented in figures and supplemented figures  are proper described , conclusions based on results are valid.     Current  protocols developed in this manuscript may improve the serological identification of HCMV 338 strains.

Some specific comment:

  1. Concentration of total DNA used in this study extracted from urine, breast milk and vaginal and saliva swabs using 77 EZ1 DSP virus kit (Qiagen, Hilden, Germany), and from blood using a QIAmp DNA mini 78 kit (Qiagen) were not presented. HCMV DNA was quantified by CMV RG PCR.  I suggest to present the concentration of total DNA in µg   as a normalization of relative copy number of HCMV DNA in each samples and indicate amount of normalized HCMV DNA as a copy/µg DNA  in multiplex assay  in 10 mcl of DNA extract .
                              1.  

Author Response

We appreciate the positive feedback of the reviewer. Regarding his suggestion, we cannot add the quantity of HCMV DNA normalized on total DNA extracted in ug, because this was not determined. However, we added the copy number of HCMV DNA/ml sample in the new Table 2.

Reviewer 2 Report

In their current manuscript, Zavaglio et al. present data on the development of an ELISA-based assay for the detection of genotype-specific antibody-responses against HCMV. While the general idea is interesting, the data shown do not confirm the feasibility.

I see the following major issue: I am not convinced about the type-specificity of the responses detected, the authors should provide more data to prove their point. The peptides derived from gB1 and gB2/3 are being detected by antibodies from patients infected by genotype gB1 or gB2/3 HCMV, respectively. However, other than these two types only gB7 induced sera is shown (which doesn't react), but not gB4, gB5, gB6 induced sera. Would these also show reactivity? Furthermore, the gB1 sera show a lot of reactivity against the gB5 and gB6 peptides, meaning that an array of these peptides would not allow for distinguishing these three genotypes. More data should be generated on the more complete cross-reactivity profile including sera induced by the other genotypes, because in the current state, alas, the data are not convincing.

Author Response

We agree with the reviewer that our analysis is not complete because we could not study infection cause by the minor gB genotypes. However, gB5, gB6 and gB7 is present in 1-2% of circulating strains. Therefore, to have the chance to identify subjects infected by these genotypes would require the collection and follow-up of hundreds of cases of primary infection, which is not feasible. Due due to the overlapping responses observed with peptides from gB1, gB5 and gB6 on one side, and gB2/3, gB4 and gB7 on the other side, our data suggests that the genotype-specific antibody response to gB could actually be divided in two groups. The first (gB1, gB5 and gB6) represents the AD169-like response and the second (gB2/3, gB4 and gB7) represents the Towne-like response. This comment has been added in the Discussion (page...). Our study was not meant to provide definitive information on genotype-specific antibody response to HCMV, but wanted to provide a further step in this topic. The search for strain specific antibody response started more than 20 years ago, and led to the identification of AD169-like and Towne-like responses. We are confident that our analysis provide a further insight due to the parallel analysis of the antibody response and infecting virus strain. This study allows the move from strain-specific response to genotype-specific (or genotype supergroup-specific) response. We acknowledge in the Discussion (page...) the limitation of our study in the lack of the analysis of the rare genotype infection. On the other hand, these genotype represent a negligible proportion of HCMV strains.

Round 2

Reviewer 2 Report

In the only slightly revised manuscript, Zavaglio et al. have added a table and one sentence in the discussion section. Neither of these minor changes addresses the major problem directly. The goal of the study was to investigate the development of a peptide-based ELISA to detect genotype-specific antibody responses to gB and gH, with the ultimate goal of using this to identify re-infections (lines 40-44, line 51-53). However, the data clearly shows that there is a lot of cross-reactivity with the tested peptides, therefore, in conclusion, the selected peptides do not allow for what the authors wanted, i.e. an ELISA that distinguishes the infecting genotypes. The authors did not / could not include data of gB4, gB5, gB6 infected patients, so the degree of cross-reactivity of antibodies by these types cannot be judged from the present data. gB7-induced antibodies were available from only one patient, which interestingly did not show reactivity to any of the peptides, including the gB7 derived ones. How the authors draw the conclusion from this that the selected peptides will work for identifying the infecting strains is not understandable to me. When it then comes to the non-primary infections, no detailed data is shown, but only bar graphs that indicate "multiple response" and "heterologous response". There is no further explanation of the reactivity pattern. 

In the present form, I don't think that the manuscript should be published, since the data just doesn't support the conclusions, and more experiments should be performed to provide a complete story. If it is too difficult to obtain sera of patients infected by the other genotypes, the authors might consider immunizing mice with the human CMV types and test the mouse sera for cross-reactivity.

Author Response

We agree with the reviewer that mice immunization with the gB5, gB6, and gB7 genotypes would provide information about the cross reactivity of the antibodies elicited by these rare genotypes. However, the aim of our study was to analyze the antibody response elicited in humans by natural infection, therefore we feel that analysis of antibodies elicited by mice immunization are outside the scope of this work.